# Multi-epitope vaccine design against Monkeypox virus: An immunoinformatics approach

Syed Ahmad[1‡], Sara Aslam[2]*, Ammara Khalid[1‡]*

1 Institute of Microbiology and Molecular Genetics, University of the Punjab, Lahore, Pakistan,
2 Department of Biomedical Engineering Technology, The Superior University, Lahore, Pakistan

‡ Equal first authors.
* Sara.aslam702@gmail.com, Sara.aslam@superior.edu.pk (SA); amara.mmg@pu.edu.pk (AK)

## Abstract

### Background

The recent outbreak of the Monkeypox virus (MPXV) across multiple regions has highlighted the need for effective vaccine. This study employed an in-silico immunoinformatics approach to design a multi-epitope vaccine against monkeypox virus.

### Methods

The IMV heparin binding surface protein (H3L) of MPV was selected as vaccine target. B and T cell epitopes were predicted and evaluated for their toxicity, allergenicity, antigenicity and global population coverage. A multi-epitope construct was assembled using adjuvant and appropriate linkers. The 3D vaccine structure was modelled, enhanced and then checked for its overall structure and physiochemical properties. The *in-silico* docking vaccine construct and TLR3 receptor was performed to check affinity and stability of vaccine-TLR3 complex. The vaccine's ability to express and activate the immune system was demonstrated by its cloning into the *E. coli* plasmid pET-28b (+).

### Results

The selected H3L protein exhibited 92.22% sequence conservation across Monkeypox virus isolates of different continents, supporting its suitability as broad-spectrum vaccine target. The molecular docking and dynamics analysis confirmed both the stability and strong binding affinity of the vaccine construct with the TLR3. Immune stimulation predicted robust humoral and cellular responses, with evidence of immunological memory formation.

**Data availability statement:** All relevant data are within the manuscript and its Supporting information files.

**Funding:** The author(s) received no specific funding for this work.

**Competing interests:** The authors have declared that no competing interests exist.

## Conclusion

Overall, we proposed a potential vaccine construct that has the potential to effectively prevents monkeypox. However, *in-vivo* and *in-vitro* investigations are needed to validate the biological effectiveness and safety of the suggested vaccine.

---

## Introduction

The outbreak of Monkeypox virus (MPXV) in several countries has sparked fears of a global pandemic. A public health emergency of international concern was declared on 14th August 2024, by the WHO when 98,001 cases of MPXV were reported in 113 countries, resulting in 184 deaths in 2024 [1]. Two clades of MPXV have been reported, of which Clade 1 is associated with a higher fatality rate. Notably, the surge in MPXV cases in 2022 is attributed to emerging virus strains exhibiting an average of 50 single nucleotide polymorphisms (SNPs). This level of genetic divergence is much higher than expected, given the estimated orthopoxvirus substitution rates [2]. To mitigate the challenge of highly virulent MPXV strains outbreak, new vaccines are needed. The traditional method of vaccine design is a lengthy and laborious process that includes culturing the pathogen and then identifying their immunogenic components. Reverse vaccinology, on the other hand, employs various bioinformatics techniques [3] that offer high specificity, safety, cost-effectiveness and the capacity to trigger humoral and cellular immune responses [4].

MPXV infects through subcutaneous portals such as nasopharynx or oropharynx, completing its life cycle involving invasion, replication and release of new viral particles inside the host cell [5]. The Monkeypox virus (MPXV) invades and replicates at the entry site prior to spreading to nearby lymph nodes and then target other organ systems in later stages of disease [6]. Cell surface proteins of MPXV play a crucial role in attachment to human cell surface receptors for invasion and viral adsorption; therefore, they serve as an important targets for vaccine development [7,8]. In VACV, H3L binds to human cell surface glycosaminoglycans and heparin sulfate while human cells deficient in glycosaminoglycans show reduced binding with viruses [9]. In another study, *in-vivo* immunization with MPXV A29L and H3L antigens induced high neutralization titers in sera against MPXV [10].

The H3L protein was chosen as the major target because of its immunodominant envelope glycoprotein that is present on the virion surface of the mature virus and is directly involved in virus-host cell attachment and membrane fusion. This orthologous gene of vaccinia and variola and has so far been illustrated to generate neutralizing antibodies and associated strong T-cell responses. The targeting of this protein is a targeted design of vaccines that minimizes the chances of off-target reactions and enhances immunogenicity. In addition to this, H3L is an early expressed protein, which demonstrates vast possibility to act as an antigen-rich area that can activate both humoral and cellular immune responses [11,12]. In another study, immunization with MPXV A29L and H3L antigens *in-vivo* showed high neutralization titers in sera

containing MPXV [13]. These studies validate the involvement of H3L in MPXV attachment and its targeting for vaccine development.

The vast majority of previous immunoinformatics studies have been done on structural proteins, including A27L, D8L or B5R, and other important viral surface proteins have had less examination. However, studies involving H3L envelope glycoprotein, a highly conserved, surface-exposed protein necessary in viral attachment and membrane fusion are limited. Antigenicity of H3L has been evidenced by experimentation that indicates robust immune neutralization in Orthopoxvirus associated diseases [14]. Functional conservation of antigens leads to a distinctive and unparalled target specificity and lessens redundancy, unlike multi-protein vaccine constructs, which is essential when concentrating on one antigen [13]. This research aims to develop a multi-epitope vaccine targeting the Monkeypox H3L protein using reverse vaccinology. This focuses on predicting potential epitopes and designing a multi-epitope vaccine to reduce viral virulence. Hence, this vaccine could be considered a potential therapeutic option against MPXV.

## Materials and methods

### Sequence retrieval and structural analysis

The genomic sequence of the IMV heparin-binding surface protein was obtained from the National Center for Biotechnology Information Database (NCBI) accession numbers NC_003310.1 (genomic) [15] and NP_536520.1 (protein). The selected protein was analyzed for its physicochemical properties using the ExPASy ProtParam tool [16], for secondary structure using the PSIPRED online tool [17], for antigenicity using VaxiJen 2.0 [17], and for allergenicity using AllerTOP v2.0 [16]. Additionally, 3D structure of IMV heparin-binding surface protein was modelled by using trRosetta https://yan-glab.qd.sdu.edu.cn/trRosetta/ [18] and THHMM v2.0 (Transmembrane Helix Prediction using Hidden Markov Models) was employed to predict the possible presence of transmembrane helices.

### B and T-cell epitope prediction and analysis

IEDB v2.0 was used to predict epitopes with an IC50 cutoff value of <200 nM, AllerTOP v2.0 and VaxiJen v2.0 to assess allergenicity, antigenicity, and related scoring parameters. The IEDB linear epitope prediction tool V2.0 [19] was used to predict linear B-cell epitopes using the default settings. Furthermore, surface accessibility analysis was performed using IEDB Bepipred online server [20]. The MHC-I and MHC-II binding prediction tools from the IEDB [21] were used for the prediction of T-cell epitopes.

The antigenicity of the chosen B and T-cell epitopes was evaluated using VaxiJen 2.0 [22]. At a threshold value of 0.35, the FASTA sequence was entered using the default setting [23]. Allergenicity and toxicity analyses were performed using AllerTop v2.0 [16] and ToxinPred2 [24], respectively.

### Computing population coverage

The population coverage of selected epitopes was calculated using IEDB's population coverage analysis tool http://tools.iedb.org/population/ with default configurations. All geographical locations of the globe were selected, and all epitopes were manually entered. Furthermore, the conservancy of the selected IMV heparin-binding protein was evaluated using the IEDB Conservancy Analysis Tool, which provided insights into the immunogenic potential of the protein.

### Vaccine construction and assemblage

From the NCBI databaseI, epitopes with the potential to elicit an immune response were gathered and linked to an adjuvant to construct a multi-epitope vaccine. The adjuvant, 50S ribosomal protein L7/L12 (UniProt ID: P9WHE3), was attached to the N-terminal of the vaccine. To facilitate protein binding and expression, a six-His tag was attached and three linkers; EAAAK, GPGPG and AAY were used to link the epitopes. This design aimed to enhance immunogenicity

and the overall immune response. Adjuvant, B-cell epitopes, MHC-I and MHC-II epitopes were manually assembled using EAAAK, GPGPG, and AAY linkers to construct a potential vaccine design.

## Evaluation different parameters of vaccine construct

The vaccine construct underwent a comprehensive evaluation of its immunogenicity, physicochemical properties, and safety. This was performed by employing AllerTop v2.0 [19], ToxinPred2 and Vaxigen v2.0 [17] for allergenicity, toxicity, and antigenicity assessments, respectively. The ExPASY-proparam tool [16] for physicochemical characteristics and SOL-pro https://scratch.proteomics.ics.uci.edu/ [25] for solubility prediction and expression analysis. The secondary structure of the multi-epitope vaccine construct were evaluated from PSIPRED online tool [17] which reveal information about α-helices, β-sheets, and coils. The tertiary structure of the vaccine was modeled using Rosetta to assess its to assess its 3D conformation and structural stability.

## Structure (3D) refinement and validation of the vaccine construct

The 3D structure of the vaccine construct was improved utilizing the GalaxyRefine web server https://galaxy.seoklab.org/refine/ [26]. The refinement process repacking and rebuilding side chains, followed by overall structural relaxation through molecular dynamics simulation. The accuracy and reliability of the structure was assessed by RAMPAGE (Ramchandran plot assessment) https://swift.cmbi.umcn.nl/servers/html/ramaplot.html and then PROCHECK https://bio.tools/procheck.

## Molecular docking and molecular dynamics simulation

The docking between TLR3 receptor's ligand binding domain (PDB ID:2AOZ) and vaccine construct was performed using ClusPro 2.0 online server [27] that predicts the binding interactions and affinities [27]. The iMODS online tool https://imods.iqf.csic.es/ [28] was used to assess molecular dynamics simulation which evaluated complex stability through torsion angle analysis. The tool offered an extensive overview of the complex's structural deformation, residue variance, eigenvalues, and RMSD values offering a comprehensive understanding of the complex's molecular interaction.

## Codon optimization and *in-silico* cloning

A two steps procedure facilitated in the promotion of expression of designed vaccine in the *E. coli* expression vector. Backtranseq (EMBOSS 6.0.1) was employed for reverse translation, and then followed by codon optimization using Java Codon Adaptation Tool (JCat) [29]. Since *E. coli* –K12 strain uses different codons from the native host, this optimization was essential for efficient expression. The construction's nucleotide sequence was submitted to the JCat server where the enzyme's cleavage sites and rho-independent transcription termination were adjusted to ensure compatibility with prokaryotic ribosome binding. The Codon Adaptation score (CAI) (1 = optimal CAI score, above 0.50 = satisfactory score) and GC content (30% to 70%) were analyzed to estimate the efficiency of expression [30]. Transcription and translation can be severely affected by values outside of this range. For the current research, the expression vector *E. coli* pET28b (+) chosen was and to assist cloning restriction sites PpuMI and DraIII were added in the construct's ends using the SnapGene tool and then the construct was amplified utilizing SnapGene *in-silico* PCR.

## Immune simulation

The C-ImmSim server (https://kraken.iac.rm.cnr.it/C-IMMSIM/index.php) was employed to simulate immune response based on epitopes and their interaction using a position-specific scoring matrix (PSSM) to check dynamics of immune response [31]. The simulation modeled cytokine kinetics where the initial IL-2 and IL-4 peak (215 days) is associated with IgM synthesis, and the secondary IL-6 and IL-10 peaks (715 days) provide support to IgG production and plasma cell differentiation. For optimized gene sequence codon adaptation index was (CAI = 0.58) and GC content was (69.13).

## Results

### Sequence retrieval and structural analysis

A full-length antigenicity analysis revealed that the IMV heparin-binding surface protein had an antigenicity score of 0.45, indicating potential as an antigen. The physicochemical properties showed aliphatic index of 97.99, suggesting thermo-stable across a broad temperature. The secondary structure prediction indicated 32% α-helices, 12% β-sheets, and 56% loops (S1 Fig). The 3D structure predicted using trRosetta (Fig 1), revealed one transmembrane helix (TMH) in a protein sequence of approximately 24.8 amino acids predicted by TMHMM. The N-terminus was likely positioned inside the cell or membrane (probability of 0.39), with residues 283–305 localized inside, 306–324 in the transmembrane helix (TMhelix), 1–282 was external to the cell or membrane (S2 Fig).

### B-cell epitopes prediction

B-cell epitopes are crucial to initiating protective immune response against viral infection tolerance/resistance. Bepipred Linear Epitope Prediction Tool (IEDB) was used to predict ten linear epitopes. Their antigenicity, toxicity, and allergenicity were evaluated using Toxinpred 2. Vaxijen 2.00, Toxinpred 2.0, and AllerTOP V2.0 correspondingly. As shown in S3 Fig and Table 1, 204–209 peptide region demonstrates potential to boost favored immunological response because of their high antigenicity score (2.15), was non-allergenicity, and non-toxic, and was selected as a strong B-cell candidate.

Using the Kolaskar and Tongaonkar method, five epitopes (47–53, 118–124, 145–151, 180–202, and 260–274) surpassed threshold value (>1.00), have ability to trigger a B-cell response and were retained (S4 Fig, Table 2). The Emini Surface Accessibility analysis identified nine epitopes (average score = 1.00, max = 6.63, min = 0.042) with seven retained based on antigenicity, toxicity, and allergenicity, analysis (S5 Fig and Table 3). The Chou–Fasman β-turn prediction (threshold = 0.93) showed peptide region ranging from 205 to 211 has a high flexible and greater ability for persuasion beta turns (average score = 0.93, max = 0.53, min = 1.37) (S6 Fig). Similarly, the Karplus and Schulz analysis (threshold = 0.98) predicted four flexible regions 15–21, 16–22, 203–209, and 158–164 (average score = 0.98, max = 1.08, min = 0.87) (S7 Fig). Based on antigenicity (> 0.35), non-toxicity, non-allergenicity, and antigenic eight B-cell epitopes

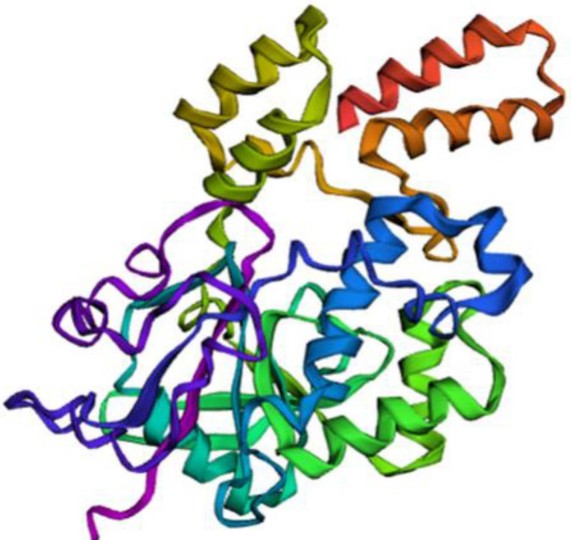

**Fig 1. The Prediction of 3D structure of the IMV heparin binding surface protein [Monkeypox virus] by trRosetta.**

**Table 1. BepiPred linear epitopes prediction using IEBD server.**

| No. | Start | End | Epitopes | Length | Antigenicity | Allergenicity | Toxicity |
|-----|-------|-----|----------|--------|--------------|---------------|----------|
| 1 | 17 | 59 | PPSETFPNVHEHINDQKFDDVKDNEVMQEKRDVVIVNDDPDH | 43 | 0.45(Antigen) | No | No |
| 2 | 67 | 78 | WTGGNIRDDDKY | 12 | 0.84 (Antigen) | Yes | No |
| 3 | 104 | 117 | LWDSKFFIELENKN | 14 | 0.99 (Antigen) | Yes | No |
| 4 | 130 | 131 | IE | 2 | – | Yes | No |
| 5 | 156 | 172 | IITGNKVKTELVIDKDH | 17 | 0.33 (Non-Antigen) | No | No |
| 6 | 204 | 209 | KSGGLS | 6 | 2.15 (Antigen) | No | No |
| 7 | 224 | 225 | KI | 2 | – | Yes | No |
| 8 | 231 | 232 | DN | 2 | – | No | No |
| 9 | 234 | 238 | AKYVE | 5 | – | Yes | No |
| 10 | 248 | 255 | RFETMKPN | 8 | 0.96 (Antigen) | Yes | No |

**Table 2. List of known epitopes predicted using Kolaskar and Tongaonkar method.**

| No. | Start | End | Peptide | Length | Antigenicity | Allergenicity | Toxicity |
|-----|-------|-----|---------|--------|--------------|---------------|----------|
| 1 | 5 | 16 | KTPVIVVPVIDR | 12 | 0.0904 (Non-Antigen) | Yes | No |
| 2 | 22 | 28 | FPNVHEH | 7 | 0.0711 (Non-Antigen) | No | No |
| 3 | 47 | 53 | RDVVIVN | 7 | 1.2994 (Antigen) | No | No |
| 4 | 59 | 66 | YKDYVFIQ | 8 | 1.0997 (Antigen) | Yes | No |
| 5 | 79 | 85 | THFFSGF | 7 | −0.4914 (Non-Antigen) | Yes | No |
| 6 | 99 | 108 | ARHLALWDSK | 10 | 0.7855 (Antigen) | Yes | No |
| 7 | 118 | 124 | VEYVVII | 7 | 0.7554 (Antigen) | No | No |
| 8 | 135 | 143 | FLRPVLKAI | 9 | −0.0988 (Non-Antigen) | No | No |
| 9 | 145 | 151 | DKKIDIL | 7 | 1.6046 (Antigen) | No | No |
| 10 | 163 | 176 | KTELVIDKDHAIFT | 14 | 0.3409 (Non-Antigen) | No | No |
| 11 | 180 | 202 | GYDVSLSAYIIRVTTALNIVDEI | 23 | 0.6234 (Antigen) | No | No |
| 12 | 210 | 217 | SGFYFEIA | 8 | 0.4608 (Antigen) | Yes | No |
| 13 | 234 | 247 | AKYVEHDPRLVAEH | 14 | 0.7234 (Antigen) | Yes | No |
| 14 | 260 | 274 | IGTVAAKRYPGVMYT | 15 | 0.6720 (Antigen) | No | No |

**Table 3. List of good surface accessibility peptides using Emini surface accessibility method.**

| No. | Start | End | Peptide | Length | Antigenicity | Allergenicity | Toxicity |
|-----|-------|-----|---------|--------|--------------|---------------|----------|
| 1 | 16 | 21 | RPPSET | 6 | −0.7503 (Non-Antigen) | Yes | No |
| 2 | 28 | 47 | HINDQKFDDVKDNEVMQEKR | 20 | 0.7627 (Antigen) | No | No |
| 3 | 54 | 61 | DDPDHYKD | 8 | −0.0068 (Non-Antigen) | No | No |
| 4 | 72 | 79 | IRDDDKYT | 8 | 0.5601 (Antigen) | No | No |
| 5 | 92 | 98 | EETKRNI | 7 | 0.1887 (Non-Antigen) | No | No |
| 6 | 113 | 118 | LENKNV | 6 | 2.0701 (Antigen) | Yes | No |
| 7 | 219 | 226 | IENEMKIN | 8 | 0.8022 (Antigen) | Yes | No |
| 8 | 232 | 240 | NSAKYVEHD | 9 | 0.2929 (Non-Antigen) | No | No |
| 9 | 247 | 256 | HRFETMKPNF | 10 | −0.4270 (Non-Antigen) | No | No |

were identified using IEDB Conservancy Analysis Tool 2.0. Efficient epitopes for eliciting B cells and were selected for the development of vaccine at 100% coverage and identity are ([204]KSGGLS[209], [145]DKKIDIL[151], [47]RDVVIVN[53], [28]HINDQKFD-DVKDNEVMQEKR[48], [118]VEYVVII[124], [260]IGTVAAKRYPGVMYT[274], [180]GYDVSLSAYIIRVTTALNIVDEI[202], and [72]IRDDDKYT[79]).

**T-cell epitopes prediction**

**MHC class-I epitopes prediction.** MHC class-I epitopes were predicted using ANN 4.0 in which Homo sapiens as a host to check broad range of MHC HLA alleles. This tool expresses the HLA-binding affinity of epitopes in terms of $IC_{50}$ (nM). Epitopes (115) were selected havig $IC_{50}$ values < 200 nM for a higher binding potential for interacting with several alleles of MHC Class-1. Of these, 36 were shortlisted based on allele interaction profile. Based on their antigenicity (>0.35), non-allergenicity, and non-toxicity, 26 epitopes were eliminated further. Allergenic or toxic epitopes which were found to have antigenic scores below 0.35 were excluded and ten were retained (S1 Table).

**MHC class-II epitopes prediction.** A total of 14 conserved predicted epitopes interacted with MHC Class-II alleles and all these epitopes had $IC_{50}$ < 200 mM. Based on the allergenicity, antigenicity, and toxicity assessment, only two out of fourteen discovered epitopes were selected for further analysis. The top binders were GFYFEIARIENEMKI and SFFGLFDINVIGLIV were identified as the top, non-allergenic binders for further analysis (S2 Table).

**Population coverage and epitope conservancy analysis**

The IEDB population coverage analysis tool was utilized to determine the global coverage of the MHC Class-I and MHC Class-II allele binding epitopes. Population coverage analysis substantial variations in the distribution of MHC HLA alleles globally. Therefore, while developing a potential operational vaccination, population coverage requires consideration. For MHC Class-I, the global population coverage was 85.85% with highest value in Europe (89.71%), followed by North America (88.11%), the West Indies (87.3%), North Africa (83.94%), Northeast Asia (83.19%), and West Africa (82.65%). On the other hand, the population coverage was lowest in Central America (3.57%). Four epitopes ILFIMFMLI, FIMFMLIFNV FTYTGGYDV and IMFMLIFNVK contributed most of the population coverage showing the global percentage to be 43.40%, 43.26%, and 43.26, 69.21%, respectively.

For MHC Class-II allele's global population coverage was 45.03%, with highest value in Europe (52.13%), followed by East Africa (50.72%), West Indies (49.72%), North Africa (49.68%), East Asia (49.39%), and South Asia (49.16%). South Africa has the lowest population coverage (5.91%). The two leading two epitopes with highest binding with MHC Class-II alleles were GFYFEIARIENEMKI & SFFGLFDINVIGLIV with global percentage 24.10% and 34.26%, respectively. Overall, the combined population coverage for multiple MHC Class I and Class II alleles reached 92.22% (S8 Fig), indicating strong global immunogenic potential of the predicted epitopes.

**Multi-epitope vaccine construction and assemblage**

For multi-epitope vaccine construct, 8 B-cell, 10 MHC Class-I, and 2 MHC Class-II epitopes were incorporated. The 50S ribosomal protein L7/L12 with (UniProt ID: P9WHE3) served as adjuvant for eliciting specific immunological reactions. The primary B-cell epitope was linked to the adjuvant at N terminus using an EAAAK linker, while GPGPG and AAY linkers were used to connect MHC-I and MHC-II epitopes. A 6x His tag for its identification and purification was added at C-terminus. The finalized construct has 485 amino acids and 51247.66 Da molecular weight. (Fig 2).

**Evaluation of the antigenicity, allergenicity and toxicity of the vaccine construct**

VaxiJen 2.0 predicted antigenicity score of 0.56 for vaccine constructs with adjuvant and without adjuvant 0.61. Regardless of whether an adjuvant is attached or not, the findings lead to the conclusion that the vaccine design is intrinsically antigenic. AllerTOP V.2 predicted that vaccine is non-allergic irrespective of adjuvant attachment. Toxinpred predicted that the vaccine design was non-toxic regardless of the adjuvant attached.

**Evaluation of the vaccine's physiochemical characteristics and solubility**

ExPASY ProtParam tool predicted the molecular weight (MW) 51384.80 Da, a theoretical pI 6.79, and instability index (II) of 27.64, indicating stability of the multi-epitope subunit vaccine (values < 40 denote stability). The aliphatic index (82.94)

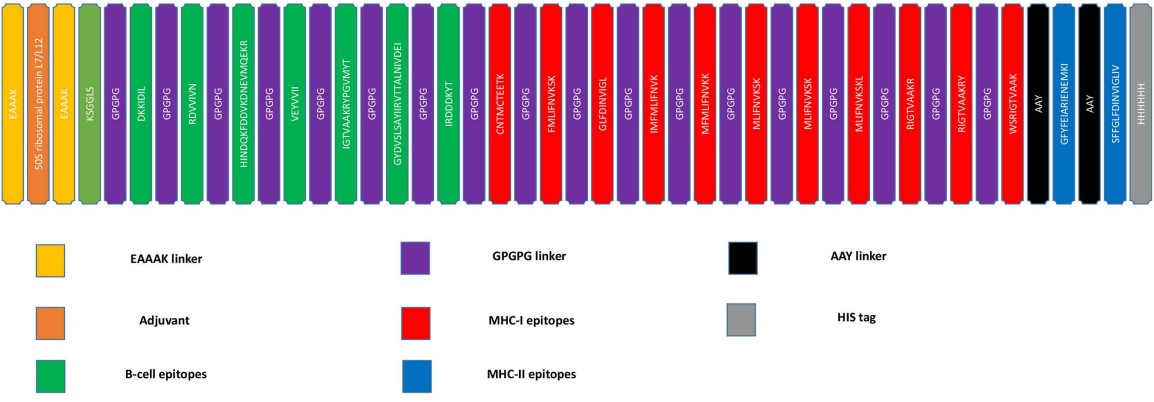

**Fig 2. Multi-epitope vaccine construct.**

suggests thermostability at high temperature, while GRAVY index (−0.207) predominantly indicates the hydrophilic nature. SOLpro3 service predicted high solubility score of 0.91 upon overexpression in *Escherichia coli*.

## Vaccine construct's secondary and structure

The secondary structure of vaccine constructed was predicted by PSIPRED tool that identified 38% coils, 10% beta strands, and 52% helix (S9 Fig). The tertiary structure was generated using TrRosetta which predicted high-confidence 3D models based on top 5 best threading templates having high coverage values. Model with highest coverage-score was selected for downstream refinement procedures (Fig 3).

## Refinement of the vaccine's 3D structure

Refinement of selected 3D model was performed using GalaxyRefine, generating five vaccine construct models (S3 Table). Model quality was assessed using multiple structural parameters including GDT-HA (90.98), Root Mean Square Deviation (0.34), MolProbity (1.13), Clash score (3.4), Poor rotamers (0.5%), and Ramachandran favored residues (99.6%). Higher GDT-HA and Ramachandran favored residues values indicated better model geometry whereas lower RMSD, MolProbity, clash score and poor rotamers reflects improve structural accuracy. Based on these assessments model 4 was chosen as the most reliable structure (S10 Fig).

## 3D structure's validation of the vaccine construct

PROCHECK server was used for the validation of refined tertiary structure. A Ramachandran plot reflecting the protein's structure was created following its structural analysis and examination (S11 Fig). Prior to refinement, 93.8% of the

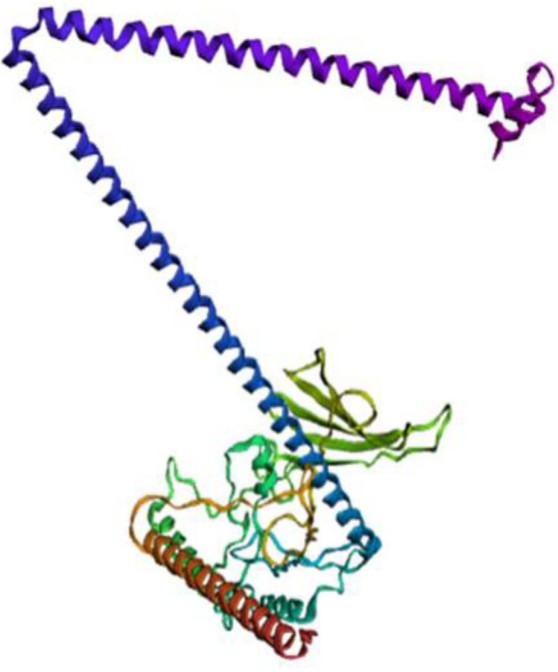

**Fig 3. 3D structure of the vaccine construct by trRosetta.**

residues were in the favored region with 5.8% in the allowed region. After refinement using PROCHECK 98.9% of residues fell in the preferred region and 1.1% of residues were in the allowed region.

## Molecular docking with TLR3 ligand-binding domain

Protein-protein docking between refined vaccine and TLR3 ligand binding domain was done using Cluspro2.0 (online). Among the 30 generated docked conformations, model number five was chosen for molecular dynamics simulation due to its minimum energy score (−1322.5 kcal/mol) and presence of 20 cluster members, indicating strong and stable binding affinity (Fig 4).

## Molecular dynamics simulation

Molecular dynamics (MD) and structural flexibility were assessed using iMODS which employs normal mode analysis (NMA) to identify macromolecular mobility and stability. iMODS does not produce time-resolved quantitative values of RMSD, RMSF, number of hydrogen bonds, binding free energies, but rather corresponds to the general dynamic behavior and stability of the structure in the form of deformability plots, B-factor graphs, covariance maps, and eigenvalue analyses. These parameters support comparative structural stability assessments and are widely accepted as computationally efficient alternatives to classical MD simulations.

The vaccine-TLR3 complex exhibited stable dynamic behavior (Fig 5). The NMA mobility plot (Fig 5a) reveals collective mobility of vaccine and identified flexible region for conformational adaptability. The B-factor NMA mobility graph (Fig 5b) evaluates residue level mobility with functional stability. The deformability graph (Fig 5c) shows segments with increased flexibility supporting internal motions. The eigenvalues (Fig 5d) indicated adequate motion stiffness required for antigenicity. The variance (Fig 5e) and covariance (Fig 5f) analysis illustrated residue coupling patterns and dynamic interactions.

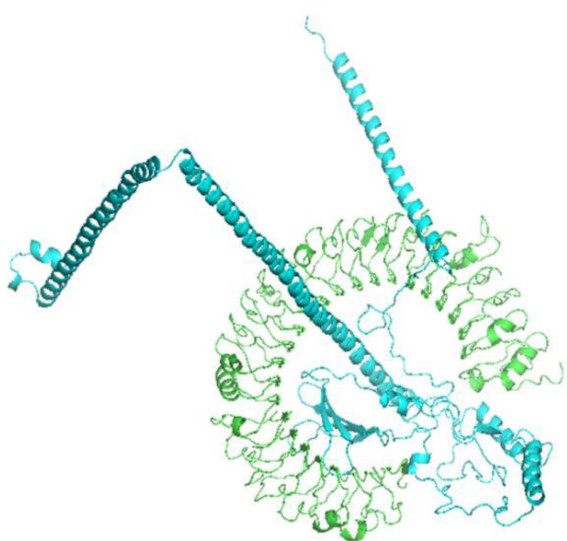

**Fig 4. Molecular docking of the vaccine with TLR3.**

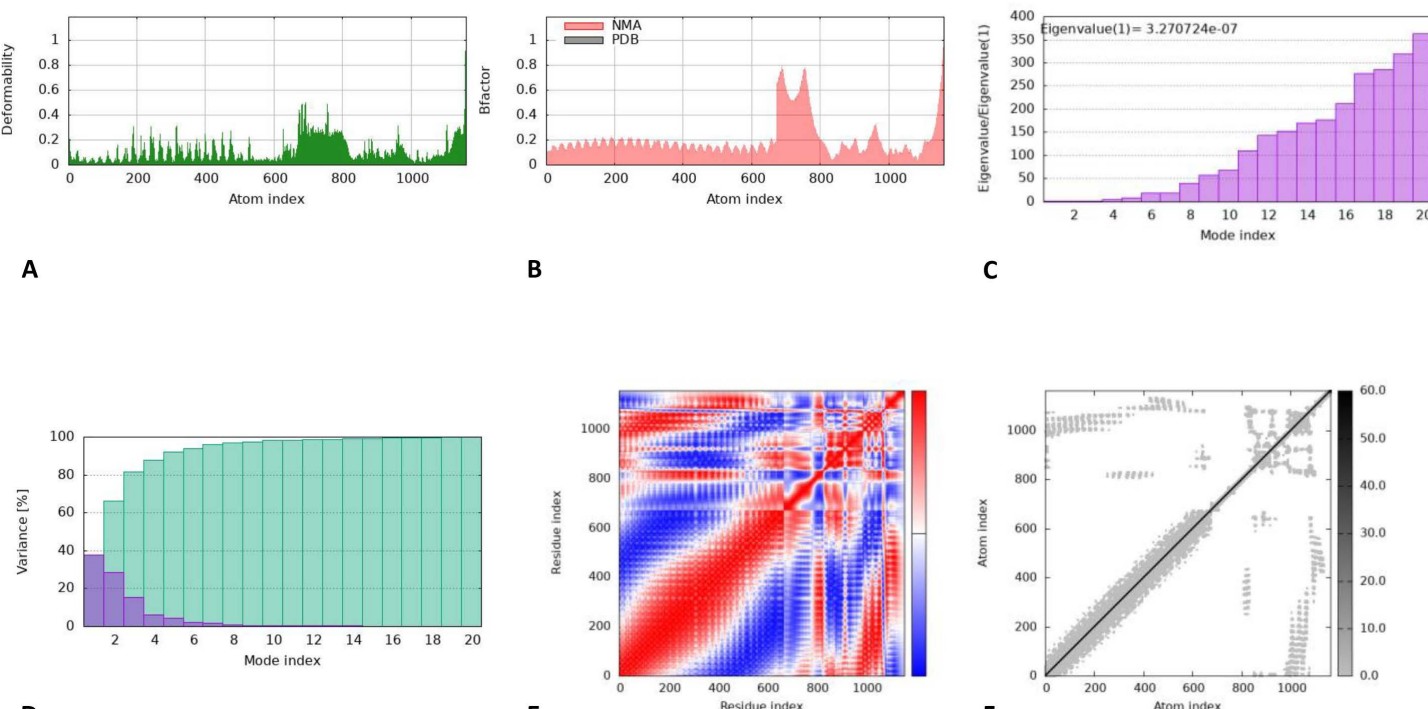

**Fig 5. Results of iMODS; a) B-factor; b) deformability plot; c) variance plot; d) eigenvalue; e) covariance matrix analysis; f) elastic network model.**

## Optimization of codons and computational cloning

Codon optimization was performed using Java Codon Adaptation Tool (JCat) and cloning was done in SnapGene. All the steps involved in cloning and codon optimization have been performed *in-silico*. Codon optimization was done to enhance expression levels in the *Escherichia coli* (K12 strain). The amino acid sequence was reverse translated using EMBOSS Backtranseq sequences to DNA sequences. Codon Adaptation Index (CAI) Plot showing your computed CAI value (0.58) compared with the optimal range of 0.8–1.0 (Fig 6).

The optimized sequence showed GC content (69.13%) within the acceptable range (Fig 7 and S12 Fig) and codon adaptation index of 0.58.

Computational cloning was carried out using SnapGene software where codon optimized sequence was inserted in pET28b (+) vector. The resulting in-silico plasmid (Fig 8) ensured heterologous cloning and subsequent expression.

## Immune simulation

*In-silico* immune simulation indicated a significant increase in primary, secondary, and tertiary immune responses (Fig 9). Fig 9B shows a steep increase in IgM titters in 5−10 days after antigen exposure to about $4 \times 10^{-4}$ titer units, indicative of the

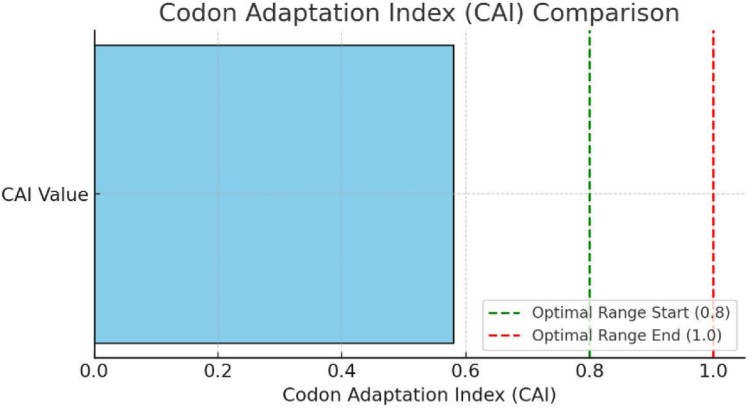

n

**Fig 6. Codon Adaptation Index (CAI) Plot showing computed CAI value (0.58) compared with the optimal range (0.8–1.0).**

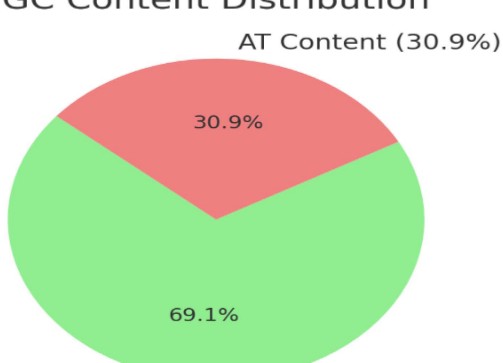

**Fig 7. GC Content distribution displaying optimized sequence composition (69.1% GC and 30.9% AT).**

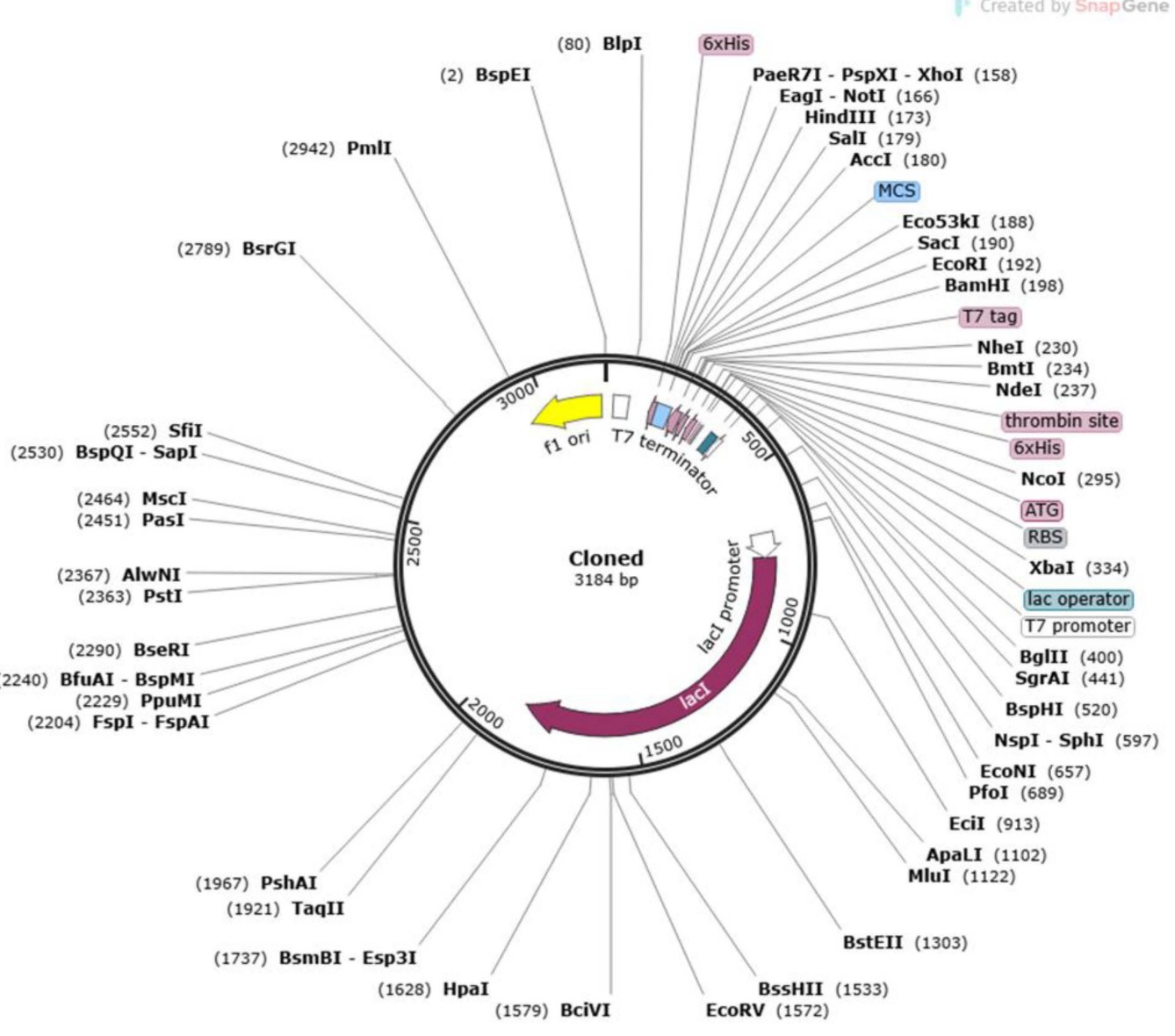

**Fig 8. SnapGene-assisted cloning of the vaccine into pET28b (+) vector.**

robust primary antibody response. IgG titers with subclasses IgG1 and IgG2 increased during days $10-25-78 \times 10^4$ titer units, reflecting a strong secondary immune response. The simulated response illustrated rise in high levels of IgG1 + IgG2, IgM, and combined IgM + IgG antibodies, accompanied by increased B-cell populations, confirming long lasting immune response. Throughout the immunization period, immunoglobulin levels rose as the antigen level fell (Fig 9A). Presence of B-cell population for longer periods of time demonstrated evidence of isotype switching and memory formation (Fig 9B, 9C). Repeated exposure of antigens led to both a decrease in T-regulatory cells (Fig 9F) and an increase in CTL and HTL T-cells (Fig 9 D, 9E). Moreover, it was predicted that the populations of dendritic and macrophage cells would remain active and grow (Fig 9 G, H). It was also predicted that there would be higher amounts of IL-2 interleukins and IFN-γ cytokines (Fig 9 M).

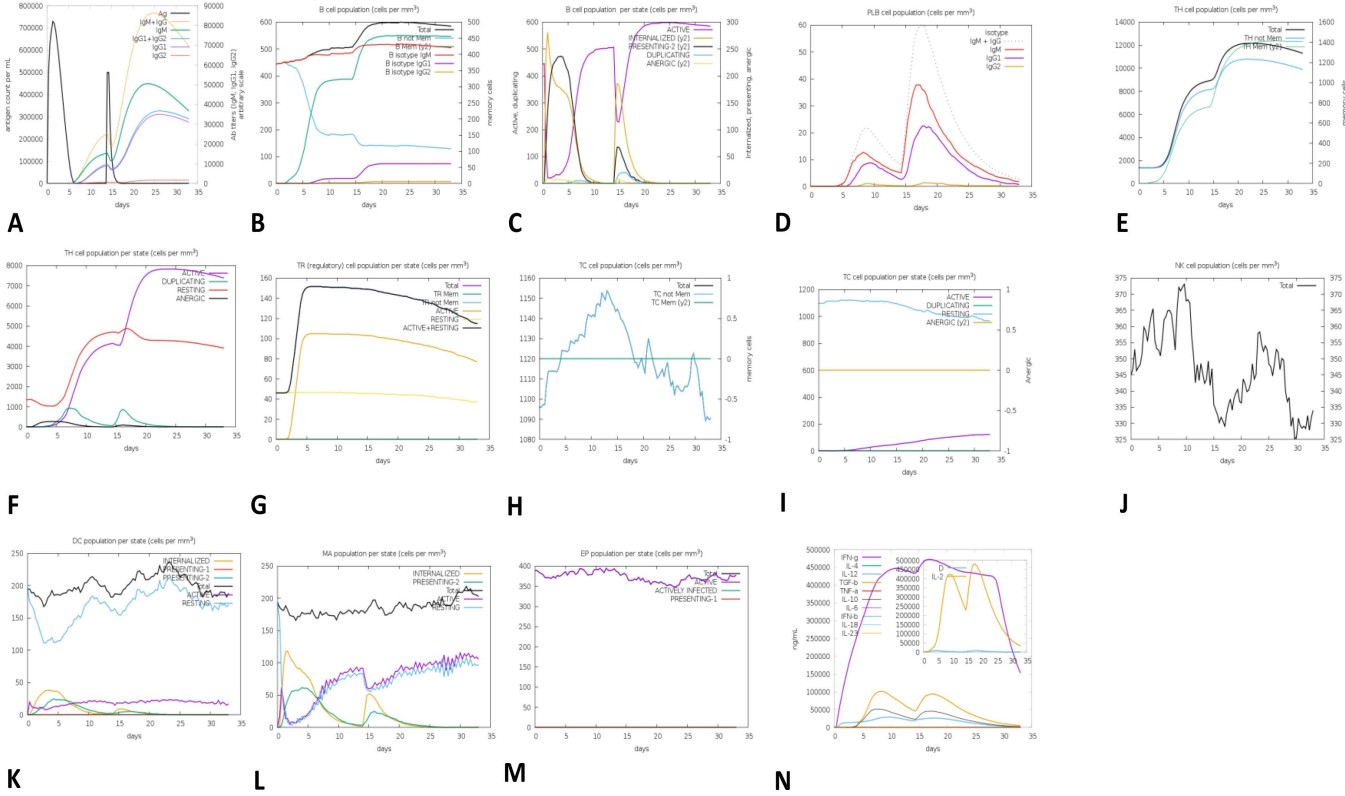

**Fig 9. C-ImmSim-assisted immune simulation predicted immune potential of the designed vaccine.** (A) Simulation showed that the injection of vaccine can raise immunoglobulin levels. The amount of antigen that is present determines the reported increase in immunoglobulin. (B) Following three doses of our vaccine design, the analysis shows that the population contains B lymphocytes (more precisely, IgM, IgG1, and IgG2). (C) The figure illustrates the diverse states of B-cell population including active, class-II presenting, internalized antigen, replicating, and antigenic ones. Different color variants are used to represent each state. (D) The figure illustrates the distribution of plasma B cells by IgM, IgG1, and IgG2 isotypes. (E) The figure shows the total number of TH cells, comprising memory cells, distinguished by IgM, IgG1, and IgG2 isotypes, post-vaccination. (F) The figure displays the amount of HTLs in both resting and active states post the administration of our vaccine, (G) The figure depicts the number of CD4 T-regulatory lymphocytes. The plot displays a representation of both total, memory, and per-entity-state counts. (H) The figure illustrates the population fluctuations of cytotoxic T cells post-vaccination within specified time intervals. (I) The graph depicts the population in two separate states resting and active cytotoxic T cells following our vaccine's administration. (J) Natural Killer Cell Population's behavior. (K) The response of the dendritic cell population to our vaccine administration, both during the active and resting phases, (L) Displays the macrophage population following MPXV immunization. (M) The EP cells are categorized into three classes: actively functioning cells, virus-affected cells, and cells displaying on class-I MHC molecules. Following repeated exposure to antigen, there were higher levels of cytokines and interleukin.

## Discussion

The escalating risk posed by the Central African clade of MPXV [32], which exhibit more fatal as compared to the most prevalent West Africa clade, underscores the urgent need of taking prevention measures. Although smallpox vaccine shows promise in cross-preventing of MPXV [33] but have related side effects such as myocarditis and pericarditis [34]. ***[35] Moreover, MPXV infection have been documented in smallpox vaccinated individuals, highlighting the need for safer and effective MPXV vaccine. The current study provides novel information because it indicates that the H3L glycoprotein is sufficient to produce a robust multi-epitope immunogenic response with high antigenicity, extensive HLA spectrum, and appreciable population coverage *in-silico*. Our H3L-derived vaccine is a simplified but effective vaccine model of a biologically crucial surface antigen unlike other computational studies that have utilized epitopes of proteins expressed by MPXV.

In this study, IMV heparin binding surface protein (H3L protein) has been selected as a target antigen due to its critical role in viral entry. H3L binds to heparin sulfate HS on the cell surface and binds specifically to the IMV membrane through its

hydrophobic C-terminus, facilitating viral entry [36]. H3L knockout in viruses showed ten times lower in viral titers and decrease infectivity [37]. Importantly, the H3L protein triggers host T and B-cell cellular immunological responses that trigger a strong IFN-γ response [38] that stimulate the release of cytokines including IL-12p70, IL-10, TNF-α, and IL-6 promoting CD8+T lymphocytes to secrete IFN-γ and proliferate, which eliminates virus-infected cells [39]. Moreover, recombinant H3L protein-immunized mice develop large titers of neutralizing antibodies (IC50=760) against VACV, showing protection against fatal viral dosages administered intranasally [40]. These features establish H3L a promising and biologically relevant target for vaccine development.

The proposed vaccine construct's is a multi-epitope vaccine design incorporating CTL-HTL-B-cell [41]. *** [42] Furthermore, as described in [43,44] we employed various linkers, such as EAAAK, AAY, and GPGPG to enhance structural stability and epitope presentation [45]. The inclusion of L7/L12 50S ribosomal protein (Locus RL7_MYCTU) (P9WHE3) further enhance immunogenicity [45].

The toll-like receptors' (TLRs) importance in eliciting immunological responses has been highlighted by earlier studies [46]. TLRs are expressed in cells which are non-immunogenic in nature like epithelial and fibroblast cells as well as in immunogenic in nature (dendritic cells and macrophages). They function as pattern recognition receptors, or PRRs using their capacity to identify conserved patterns PAMPs (pathogen-associated molecular patterns). These PAMPs originate from various microorganisms which has a crucial function in innate immunity [47].

Molecular docking of vaccine and TLR-3 showed strong binding affinity with the TLR-3 residues and vaccine constructs, suggesting efficient innate immunogenic reactions. The structural stability of TLR3-vaccine complex reported in our study is in consistent with previous findings [48]. High titers of neutralizing antibodies and cytokines after vaccination were identified in *in-silico* immunological modeling. These findings suggest that our vaccine construct is effective and triggers stronger immunological response which is in turn capable of providing protection against MPX disease.

Several recent immunoinformatics studies have proposed MPXV multi-epitope vaccine that have either experimentally non-validated polyvalent antigen panel [49–52] or broad membrane glycoprotein focus [53]; however these studies have the limitations that they often emphasize epitope prediction without extensive structural refinement, receptor-level interactions analysis or practical expression planning of vaccine construct. In contrast our study utilizes experimentally validated H3L antigen that is mechanistically associated with viral entry and neutralization, incorporates more comprehensive orthogonal quality checks (structure quality, dynamics, immune kinetics, and cloning readiness). While earlier studies reported promising antigenicity, our vaccine construct demonstrate superior population coverage and complete single conserved H3L antigen that simplifies downstream antigen production as compared to polyvalent design. Even though this work gives broad computational predictions, *in-silico* models cannot reproduce biological complexity, such as immune regulation, protein expression and folding. Thus, *in-vitro* and *in-vivo* validation should be considered as the next steps to ensure the immunogenicity, safety, and efficacy of the proposed vaccine construct.

## Conclusion

The present study established the basis for a broad-spectrum vaccine candidate by employing innovative reverse vaccinology and immunoinformatics approaches to target the intracellular mature virions (IMV) heparin binding surface protein of MPXV. The molecular dynamics simulation analysis confirmed both the stability and strong binding affinity of the vaccine construct with the TLR3 complex, indicating its potential effectiveness through induction of strong humoral and cellular immunological responses against MPXV *in-silico*. To validate the findings of the current study a thorough experimental validation through preclinical research is necessary to guarantee the safety and effectiveness of the vaccine candidate.

## Supporting information

**S1 Fig. Secondary structure prediction of IMV heparin binding surface protein [Monkeypox virus] by PSIPERED analysis.**
(DOCX)

**S2 Fig. Transmembrane helix prediction in proteins by TMHMM.**
(DOCX)

**S3 Fig. BepiPred linear epitopes predicted via IEBD analysis resource. Threshold line: anything above it is predicted as an epitope.**
(DOCX)

**S4 Fig. Antigenicity analysis by Kolaskar and Tongaonkar method.**
(DOCX)

**S5 Fig. Antigenicity analysis by Emini surface accessibility method.**
(DOCX)

**S6 Fig. Prediction of beta turns in IMV heparin binding surface protein by using Chou and Fasman beta turn analyzing algorithm.**
(DOCX)

**S7 Fig. Karplus and Schulz flexibility prediction for the flexibility of IMV heparin binding surface protein.**
(DOCX)

**S8 Fig. Global population coverage of selected MHC class I, II & combined MHC class I & II epitopes.**
(DOCX)

**S9 Fig. Extrapolation of secondary structure by PSIPRED tool.**
(DOCX)

**S10 Fig. Model 4 of the vaccine construct by GalaxyRefine.**
(DOCX)

**S11 Fig. Ramchandran Plot of Model 4 of the vaccine construct by PROCHECK server.**
(DOCX)

**S12 Fig. CodonUsage adapted to Escherichia coli (strain K12) using Jcat.**
(DOCX)

**S1 Table. Most potential non-allergen, nontoxic, 10 T-cell epitopes with interacting MHC-I alleles.**
(DOCX)

**S2 Table. Most potential non-allergen, nontoxic, 2 T-cell epitopes with interacting MHC-II alleles, epitope conservancy score, $IC_{50}$ less than 200 and percentile rank.**
(DOCX)

**S3 Table. Five refined models of the vaccine construct by GalaxyRefine.**
(DOCX)

## Author contributions

**Conceptualization:** Syed Ahmad.

**Data curation:** Syed Ahmad, Sara Aslam.

**Formal analysis:** Syed Ahmad.

**Funding acquisition:** Syed Ahmad.

**Investigation:** Sara Aslam.

**Methodology:** Syed Ahmad.

**Project administration:** Ammara Khalid.

**Resources:** Ammara Khalid.

**Supervision:** Ammara Khalid.

**Validation:** Sara Aslam, Ammara Khalid.

**Visualization:** Sara Aslam, Ammara Khalid.

**Writing – original draft:** Syed Ahmad, Sara Aslam.

**Writing – review & editing:** Sara Aslam, Ammara Khalid.

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
