## [Decision Letter · Decision Letter 0]

29 Sep 2025

Dear Dr. Aslam,

We look forward to receiving your revised manuscript.

Kind regards,

Syed Nisar Hussain Bukhari

Academic Editor

PLOS ONE

Journal Requirements:

Reviewers' comments:

Reviewer's Responses to Questions

**Comments to the Author**

1. Is the manuscript technically sound, and do the data support the conclusions?

Reviewer #1: No

Reviewer #2: Yes

Reviewer #3: Yes

2. Has the statistical analysis been performed appropriately and rigorously?

Reviewer #1: No

Reviewer #2: I Don't Know

Reviewer #3: Yes

3. Have the authors made all data underlying the findings in their manuscript fully available?

Reviewer #1: Yes

Reviewer #2: Yes

Reviewer #3: Yes

4. Is the manuscript presented in an intelligible fashion and written in standard English?

Reviewer #1: Yes

Reviewer #2: No

Reviewer #3: Yes

Reviewer #1: The study addresses an important global health problem and applies an immunoinformatics pipeline to design a multi-epitope vaccine candidate against Monkeypox virus. However, the current version requires substantial improvements before it can be considered for publication.

Major Comments:

1. Several similar computational vaccine studies for MPXV have already been published. Please clarify how this study differs and contributes new insights. A comparison with prior MPXV vaccine design papers would strengthen the work.

2. The rationale for selecting only the H3L protein should be better justified, with supporting evidence of its immunogenic role.

3. The manuscript does not provide sufficient details on prediction tools, databases, thresholds, and simulation parameters. For reproducibility, please include exact versions, cutoffs, and settings.

4. Claims of stability and strong binding must be supported with quantitative data (RMSD, RMSF, hydrogen bond numbers, binding free energies). Figures and tables should be included.

5. Please provide quantitative readouts (antibody titers, cytokine levels) rather than qualitative statements.

6. Clarify that cloning was performed computationally. Include codon optimization indices and GC content with visuals.

7. The conclusion should more explicitly state that the study is an in silico prediction only and requires experimental validation.

Minor Comments:

1. The abstract should include quantitative results rather than vague phrases.

2. Correct the keyword “immunoinformatic” to “immunoinformatics.”

3. Improve grammar (e.g., “has emphasize” → “has emphasized”).

4. Ensure recent references (2022–2024) on MPXV vaccine design are cited.

Overall: The manuscript has potential but needs major revision to meet PLOS ONE’s standards of methodological rigor, reproducibility, and clarity.

Reviewer #2: This article is solid in context and contributes valuable insights; however, several areas require refinement to enhance clarity and precision.

Abstract: The reference to 92.22% needs to be expressed more clearly. It is not immediately evident what “worldwide” refers to in this context. Please clarify.

Methods: It is unclear whether the cloning and testing were performed directly by the authors or whether these steps refer to previously published work. This distinction should be explicitly clarified to avoid ambiguity.

Results vs. Methods: Some descriptions in the Results section overlap with methodological details (e.g., the subsection on MHC Class II epitope prediction).

Numerical Consistency: The manuscript would benefit from consistent rounding of numerical values, ideally to two or three decimal places, to improve accuracy and readability.

Conclusion: The current conclusion is overly general. A more specific articulation of the study’s contribution and implications is needed. For example, highlight how the findings advance vaccine design, what novel contributions were made compared to existing approaches, and what the next steps for validation should be.

Overall: This is a complex and ambitious research study.

Ethical considerations: The study appears to have been conducted using computer-aided vaccine prediction methods. No major concerns were identified regarding dual publication, research ethics, or publication integrity.

The authors employed multiple comparative approaches, simulations, and iterative refinements to assemble a final predictive vaccine model. While this breadth adds value, it also contributes to the manuscript’s length. Rationalising text and focusing on the most critical findings could improve readability and impact.

Reviewer #3: The study addresses an important topic and follows standard computational pipeline. It has several strengths including:

- The topic is highly relevant given the ongoing concerns about Monkeypox virus outbreaks.

- The manuscript is well structured and covers the full computational pipeline.

- The choice of the H3L protein is reasonable and supported by prior literature.

However, it has certain areas for improvement including:

- The authors study should clarify what is new compared to other recent MPXV vaccine design papers using computational approaches.

- The manuscript requires editing for grammar and clarity to improve readability.

- The discussion and conclusion should place stronger emphasis on the limitations of in silico approaches.

**Do you want your identity to be public for this peer review?** For information about this choice, including consent withdrawal, please see our Privacy Policy

Reviewer #1: No

Reviewer #2: **Yes:** Dr Isse Ali

Reviewer #3: No

---

## [Author Response · Author response to Decision Letter 1]

9 Dec 2025

Multi-epitope vaccine design against Monkeypox virus: An immunoinformatic approach

Dear Editor/reviewers,

We thank reviewers for their time and consideration in providing valuable feedback on our manuscript. All authors really appreciate comments, carefully addressed all comments and the revised manuscript includes the necessary changes. Below, we provide detailed responses to each comment with references to the updated sections in the manuscript. We believe these revisions have significantly improved the quality of the manuscript.

Thanks again,

Dr. Sara Aslam

Reviewer 1

Comment 1: Several similar computational vaccine studies for MPXV have already been published. Please clarify how this study differs and contributes new insights. A comparison with prior MPXV vaccine design papers would strengthen the work.

Response: Thank you for pointing out this gap. In this revised manuscript, we have provided rationale in the discussion section (highlighted in yellow).

Comment 2: The rationale for selecting only the H3L protein should be better justified, with supporting evidence of its immunogenic role.

Response: This justification has been incorporated into the Introduction section of the revised manuscript (highlighted in yellow), supported by additional recent references.

Comment 3: The manuscript does not provide sufficient details on prediction tools, databases, thresholds, and simulation parameters. For reproducibility, please include exact versions, cutoffs, and settings.

Response: To provide maximum transparency and to facilitate reproducibility, we now have provided detailed methodological information in the Materials and Methods section. In particular, names, database sources, software version, and threshold values of every step of the analysis process, including epitope prediction, antigenicity testing, allergenicity screening, molecular docking, and immune simulation have been given in both methodology and results section. These revisions are added throughout the manuscript, thereby ensuring that future researchers can replicate and validate our approach.

Comment 4: Claims of stability and strong binding must be supported with quantitative data (RMSD, RMSF, hydrogen bond numbers, binding free energies). Figures and tables should be included.

Response: The computer-generated molecular dynamics (MD) of the present study was conducted with the help of the iMODS online server, and it uses normal mode analysis (NMA) to make predictions of the structural flexibility and stability of the vaccine receptor complex. We observed that iMODS does not produce time-resolved quantitative values of RMSD, RMSF, number of hydrogen bonds, binding free energies, but rather corresponds to the general dynamic behavior and stability of the structure in the form of deformability plots, B-factor graphs, covariance maps, and eigenvalues. The NMA results are relevant fort the estimation of relative structural stability and be applied as a computationally robust and reliable source to traditional MD trajectories in in-silico vaccine studies. We have clarified this limitation and explanation in the Results section.

Comment 5: Please provide quantitative readouts (antibody titers, cytokine levels) rather than qualitative statements.

Response: Based on the in-silico immune simulation results, quantitative representation of the immune response has been incorporated. For the dynamics of antibody titer, Figure No 7B shows a steep increase in IgM titer in 5 -10 days after antigen exposure to about 4 10 -4 titer units, which is the primary antibody response. Later, IgG titers with subclasses IgG1 and IgG2 elevate during days 1025 to 78x 104 titer units, which is a strong secondary immune response. These quantitative values are now discussed in the Results section, offering a detailed study of the immune response strength and time duration.

Comment 6: Clarify that cloning was performed computationally. Include codon optimization indices and GC content with visuals.

Response: The simulation does not directly plot the cytokines, but it is determined by using antibody kinetics, where the initial IL-2 and IL-4 peak (215 days) is associated with IgM synthesis, and the secondary IL-6 and IL-10 peaks (715 days) provide support to IgG production and plasma cell differentiation. We have made clear that no cloning was done experimentally, only computationally (in silico). We have also added the codon adaptation index (CAI = 0.58) and GC content (69.135) of the optimized gene sequence to consider the expression possibilities in E. coli (strain K12).

These additions appear on the revised manuscript.

Here are the visuals you requested:

Figure 8: Codon Adaptation Index (CAI) Plot showing your computed CAI value (0.58) compared with the optimal range (0.8–1.0).

Figure 7: GC Content Distribution displaying your optimized sequence composition (69.1% GC and 30.9% AT).

Comment: The abstract should include quantitative results rather than vague phrases.

Response:

Comment: Correct the keyword “immunoinformatic” to “immunoinformatics.”

Response: Thank you for identifying the issue, it has been corrected throughout the manuscript.

Comment: Improve grammar (e.g., “has emphasize” → “has emphasized”).

Response: In revised version of this manuscript, this word is eliminated.

Comment: Ensure recent references (2022–2024) on MPXV vaccine design are cited.

Response: Recent reference (2022–2024) on MPXV vaccine design are added in the manuscript.

Reviewer 2

Comment 1: Abstract: The reference to 92.22% needs to be expressed more clearly. It is not immediately evident what “worldwide” refers to in this context. Please clarify.

Response: The 92.22% global was used to refer to the conserved sequence of the H3L protein among the MPXV isolates found globally. It has been revised in manuscript.

Comment 2: Methods: It is unclear whether the cloning and testing were performed directly by the authors or whether these steps refer to previously published work. This distinction should be explicitly clarified to avoid ambiguity.

Response: All the steps involved in cloning and codon optimization have been performed in-silico by the authors. Computational cloning was done in SnapGene, codon optimization in the Java Codon Adaptation Tool (JCat). The optimized construct had Codon Adaptation Index (CAI) of 0.58 and GC content of 69.13%, which showed considerable expressive capabilities in E. coli K12.

Comment 3: Results vs. Methods: Some descriptions in the Results section overlap with methodological details (e.g., the subsection on MHC Class II epitope prediction).

Response: Redundant methodological information (e.g. description of MHC-I epitope prediction tool) has been repositioned in the Methods section as opposed to the Results since this section is more organized and makes sense.

Comment 4: Numerical Consistency: The manuscript would benefit from consistent rounding of numerical values, ideally to two or three decimal places, to improve accuracy and readability.

Response: In the manuscript, all of the numerical values have been standardized to two decimal points, which is consistent and clear. These can be

“92.2231% → 92.22%” and “CAI = 0.5798 → CAI = 0.58”.

Comment 5: Conclusion: The current conclusion is overly general. A more specific articulation of the study’s contribution and implications is needed. For example, highlight how the findings advance vaccine design, what novel contributions were made compared to existing approaches, and what the next steps for validation should be.

Response: Thank you for identifying this issue. Conclusion is updated.

Reviewer 3

Comment 1: The authors study should clarify what is new compared to other recent MPXV vaccine design papers using computational approaches.

Response: Thank you for the comment. We have added the comparison in the introduction and discussion section (highlighted in yellow).

Comment 2: The manuscript requires editing for grammar and clarity to improve readability.

Response: The entire manuscript has undergone a grammatical review to enhance precision and readability.

Comment 3: The discussion and conclusion should place stronger emphasis on the limitations of in-silico approaches.

Response: Thank you for identifying the missing aspect. We have included the limitations of in-silico approaches.

---

## [Decision Letter · Decision Letter 1]

19 Jan 2026

Multi-epitope vaccine design against Monkeypox virus: An immunoinformatic approach

PONE-D-25-35744R1

Dear Dr. Aslam,

We’re pleased to inform you that your manuscript has been judged scientifically suitable for publication and will be formally accepted for publication once it meets all outstanding technical requirements.

Kind regards,

Syed Nisar Hussain Bukhari

Academic Editor

PLOS One

Additional Editor Comments (optional):

Reviewers' comments:

Reviewer's Responses to Questions

**Comments to the Author**

Reviewer #2: All comments have been addressed

Reviewer #3: All comments have been addressed

2. Is the manuscript technically sound, and do the data support the conclusions?

Reviewer #2: Yes

Reviewer #3: Yes

3. Has the statistical analysis been performed appropriately and rigorously?

Reviewer #2: Yes

Reviewer #3: Yes

4. Have the authors made all data underlying the findings in their manuscript fully available?

Reviewer #2: Yes

Reviewer #3: Yes

5. Is the manuscript presented in an intelligible fashion and written in standard English?

Reviewer #2: Yes

Reviewer #3: Yes

Reviewer #2: The authors have addressed the review comments precisely, and this is a solid paper. The developed vaccine demonstrates broad global coverage based on detailed predictive models supported by appropriate statistical methods, and it shows strong conservation across a range of MPXV strains.

Reviewer #3: - The novelty was duly clarified.

- Grammatical precision issue was addressed.

- Limitations of in-silico approach were mentioned.

All the issues were hence addressed.

**Do you want your identity to be public for this peer review?** For information about this choice, including consent withdrawal, please see our Privacy Policy

Reviewer #2: **Yes:** Dr Isse Ali

Reviewer #3: No

---

## [Editor Report · Acceptance letter]

PONE-D-25-35744R1

PLOS One

Dear Dr. Aslam,

I'm pleased to inform you that your manuscript has been deemed suitable for publication in PLOS One. Congratulations! Your manuscript is now being handed over to our production team.

Kind regards,

on behalf of

Dr. Syed Nisar Hussain Bukhari

Academic Editor

PLOS One